# Sociodemographic Correlates of Obesity among Spanish Schoolchildren: A Cross-Sectional Study

**DOI:** 10.3390/children7110201

**Published:** 2020-10-28

**Authors:** José Francisco López-Gil, Alba López-Benavente, Pedro Juan Tárraga López, Juan Luis Yuste Lucas

**Affiliations:** 1Departamento de Actividad Física y Deporte, Facultad de Ciencias del Deporte, Universidad de Murcia (UM), San Javier, 30720 Murcia, Spain; 2Departamento de Expresión Plástica, Musical y Dinámica, Facultad de Educación, Universidad de Murcia (UM), Espinardo, 30100 Murcia, Spain; alba.lopez3@um.es (A.L.-B.); jlyuste@um.es (J.L.Y.L.); 3Departamento de Ciencias Médicas, Facultad de Medicina, Universidad Castilla-La Mancha (UCLM), 02008 Albacete, Spain; pjtarraga@sescam.jccm.es

**Keywords:** adiposity, waist circumference, anthropometry, children, socioeconomic factors, health inequality indicators

## Abstract

Some studies have been conducted in order to assess the association between weight status (assessed by body mass index) and socio-demographic factors. Nevertheless, only a few of them have indicated these associations by other anthropometric parameters (e.g., skinfolds). The aim of this study was to determine, compare, and examine the influence of age, sex, type of the schooling, per capita income, area of residence, and immigrant status on obesity parameters in schoolchildren aged 6–13 from the Region of Murcia. A cross-sectional study was carried out in six different Primary schools of the Region of Murcia (Spain). A total sample of 370 children (166 girls) aged 6–13 (8.7 ± 1.8) were selected. In order to determine participants’ body composition, body mass index, waist circumference, waist-to-height ratio, and skinfold measurements were calculated. Higher associations of excess of weight (OR = 1.96; 95%CI = 1.19–3.20) and abdominal obesity (OR = 3.12; 95CI% = 1.49–6.94) were shown in the case of children from public schools. A greater association of high trunk fat mass was found in children from municipalities with high per capita income (OR = 3.20; 95%CI = 1.05–9.77). Therefore, lower association of having an inadequate %BF was found in the participants aged 6–9 (OR = 0.38; 95%CI = 0.24–0.54), and immigrant students (OR = 2.63; 95%CI = 1.69–4.10). Our study suggested that overweight/obesity among schoolchildren in the Region of Murcia is higher than the overall prevalence of Spain. The results of the adjusted analyses showed that age, type of schooling, per capita income, and immigrant status were associated with obesity parameters.

## 1. Introduction

Globally, overweight and obesity have reached worryingly high levels [1], especially in Europe, where this situation represents a major public health threat [2]. Thus, the last World Health Organization (WHO) European Childhood Obesity Surveillance (COSI) study has ranked Spain with the highest obesity prevalence rate in Europe (17.7%) [2]. The risk of morbidity and mortality in adult life increases in those who are overweight or obese (understood as excess weight) in their childhood or adolescence [3], which also increases the likelihood of having worse risk parameters for cardiovascular disease [4]. This is why the WHO Member States have committed to ensuring that childhood obesity figures do not increase by 2025 [5].

Regarding obesity identification, body mass index (BMI) is the most applied anthropometric parameter in children; existing several reference cut-off points, both internationally [6,7,8] and nationally [9]. However, even though BMI can be an assessment tool to help detect overweight or obese children, it has been advised that it does not identify the majority of children who will develop morbidities related to obesity in adult life [10]. Also, there is a great variability in the association of BMI and fat mass, which could be due to physiological changes, the level of pubertal maturation, and the sex of the children [11]. In the same way, during the process of growth and development, several changes occur in body composition, mainly in the storage and distribution of fat, muscle, and bone mass according to age and sex [12]. Thus, a recent systematic review performed by Orsso et al. [13] has suggested that skinfold measurements are one of the possible reliable methodologies for determining body composition in paediatric obesity. At the same time, Schröder et al. [14] found a large proportion of normal-weight Spanish children who present abdominal obesity. For these reasons, evaluation accompanied by other body composition variables is highly recommended, such as waist-to-height ratio (WHtR), waist circumference (WC), or skinfolds in order to effectively determine the risk in participants [15].

Following this line, some studies have been conducted in order to assess the association between weight status (assessed by BMI) and socio-demographic factors, such as sex [16], type of schooling [17], area of residence [18], immigrant status [19], or per capita income [20]. For example, children from public schools have been associated with a higher prevalence of overweight or obesity [17], as well as children belonging to an immigrant family [21]. Equally, the prevalence of obesity is significantly higher in school children from families with lower incomes compared to those with higher ones in Spain [22], being so in the case of children from rural areas [23]. Hence, some of these studies have shown certain degree of association between these variables. Notwithstanding, only a few of them have indicated these associations by other anthropometric parameters (e.g., skinfolds) [24], and there is a gap of knowledge in this sense.

In addition to this, children and adolescents who have obesity are about five times as likely to be obese in adulthood than their counterparts who do not present obesity [25]. Thus, because of the long-term implications of childhood obesity and the influence on their health in adulthood, studies to clarify the relationship between sociodemographic characteristics and the risk of overweight and obesity merits further consideration [26]. As a result, identifying those children at high risk of being obese in adulthood and its related factors could be a useful matter; particularly in the Region of Murcia, where the greatest figures of excess weight children has been pointed out [27].

According to the worrying situation of childhood obesity in Spain, particularly in the Region of Murcia, as well as the lack of information about those sociodemographic factors that could be related to childhood obesity, the aim of this research was to determine, compare and examine the influence of age, sex, type of the schooling, area of residence, immigrant status, and per capita income on obesity parameters in schoolchildren aged 6–13 from the Region of Murcia.

## 2. Materials and Methods

### 2.1. Sample

This study had a cross-sectional design. All nine primary schools from the Valley of Ricote (Region of Murcia, Spain) were invited to take part, with six schools agreeing to participate. Out of 972 possible participants, a total of 370 schoolchildren (166 girls and 204 boys) with ages between 6–13 years old (8.7 ± 1.8) participated, representing a response rate of 38.1%. For this study we used the non-probability sampling technique. In spite of this, all schoolchildren were likewise offered to join.

All the children involved in the study had to provide a signed consent form by their parents/legal guardians. Before the performance of the study, both parents and children were informed about the purpose of the research and the type of tests that would be carried out. Data was obtained in the 2017/2018 academic year. All the measurements were performed during Physical Education classes. As an inclusion criterion, only schoolchildren aged 6–13 who had obtained positive informed consent from their parents or legal guardians were included.

Regarding the origin of the immigrant students (*n* = 71), 61.1% are of South American origin, 29.1% of Arab origin, 5.6% from other European countries, 2.8% born in Asia, and 1.4% of African origin.

The Bioethics Committee of the University of Murcia (ID 2218/2018) approved the present study. It has been executed respecting the human rights of the participants and following the Helsinki Declaration.

### 2.2. Data Collection

Sex and age were self-reported. The type of schooling was dichotomised into two categories: (1) public and (2) private with public funds. Area of residence was divided into (1) urban (>5000 inhabitants) and (2) rural (≤5000 inhabitants) [28]. Students who met at least one of the following conditions were considered to be immigrant: (a) born outside Spain, (b) immigrant parents, or (c) at least one of their parents comes from another country. The socio-economic characteristics were determined by the per capita income of the different municipalities. Thus, this variable was dichotomised into two groups: (1) high per capita income (€20,000 or more) and (2) low per capita income (under €20,000). A portable height rod with an accuracy of 0.1 cm (Leicester Tanita HR 001, Tokyo, Japan) was used so as to determine participants’ height. Their bodyweight was measured with an electronic scale with an accuracy of 0.1 kg (Tanita BC-545, Tokyo, Japan). BMI was calculated from the ratio between body weight (kg) and the height squared of the participants (m^2^). Besides, weight status was determined using the age-specific and sex-specific thresholds provided by the WHO [6]. WC was measured with a precision of 0.1 cm at the level of the intersection between the last rib and the border of the iliac crest, using a constant tension tape. Children were split by “no high trunk fat mass” and “high trunk fat mass” according to sex- and age-specific cut-off values [29]. Moreover, the WHtR was calculated in order to determine the prevalence of abdominal obesity and children were classified as “no abdominal obesity” and “abdominal obesity” [30]. Skinfold measurements (subscapular, iliac crest, biceps and triceps) were taken using pre-calibrated steel callipers with a precision of 0.2 mm (Holtain, Crosswell, Crymych, United Kingdom) and following the guidelines of the International Society for the Advancement of Kinanthropometry (ISAK) [31]. To calculate the body density, the log of the sum of skinfolds measurements was considered [32]. Siri formula was used in order to compute body fat from body density [33], and fat-free mass was then determined minus total body mass and body fat (BF) mass. Likewise, participants were divided into “adequate adiposity” and “high adiposity” [34].

All the measurements were performed by the same trained researcher. Two measurements were taken and, if existed discrepancies exceeding 0.1 centimetres (cm) between measurements the procedure, was performed again. The relative technical error of measurement (TEM) was obtained by performing a number of repeated measurements on the same participant, taking the differences and inserting them into a correct equation. For intra-observer TEM of the two measurements taken the following equation was applied: √ΣD/2N, where “D” is the difference between measurements and “N” is the number of participants measures. Also, the absolute TEM was multiplied by 100 and divided by the variable average value in order to provide the relative TEM (%TEM). The %TEM were 0.3%, 0.2%, 0.5%, and 2.3%, for weight, height, WC, and skinfolds, respectively.

### 2.3. Data Management and Statistical Analysis

For all continuous variables, data was shown as mean (M) and standard deviation (SD). Conversely, all categorical variables were expressed as number (*n*) and percentage (%). Likewise, the confidence intervals (95%) were calculated. To assess the differences between groups of age, sex, type of schooling (public/private), per capita income (high/low), area of residence (urban/rural), and immigrant status (native/immigrant), the Chi-squared test was used or, if expected values were lower than 5, Fisher’s Exact test was applied. Furthermore, binary logistic regression analyses were performed to determine the association between sociodemographic characteristics and the different obesity parameters. Finally, data analysis was performed using the software SPSS (IBM Corp, Armonk, NY, USA) for Windows (v.24.0). Statistical significance was maintained at *p*-value ≤ 0.050.

## 3. Results

Data of age and continuous variables of anthropometric characteristics are shown in Table 1, as well as, the prevalence according to different obesity parameters are presented in Table 2. According to BMI, the prevalence of excess weight was 52.4% (WHO criteria). Moreover, when WC was considered, the values ranged from 14.6% to 20.3%, for abdominal obesity and high trunk mass fat, respectively. Conversely, a 45.4% presented high adiposity based on %BF.

Figure 1 shows the differences between the prevalence of obesity parameters according to the sociodemographic factors analysed. In relation to sex, boys had higher excess weight (43.6%) than girls (42.2%), with no statistically significant differences found. This absence of statistically significant differences was also shown in the case of WC, WHtR and %BF. On the other hand, no statistically significant differences were obtained for BMI, WC, and WHtR with regard to age group; not being so in the case of %BF (*p* < 0.001). So, a greater number of children aged 10–13 with great levels of %BF (22.3% higher) was identified. Regarding the type of schooling, children from public school showed greater values in all variables of obesity. Likewise, statistically significant differences were shown in most of obesity parameters that were examined (except for %BF). Also, higher prevalence of excess weight, abdominal obesity, high trunk fat mass, and high adiposity was found in the case of high per capita income group (only statistically significant for BMI). Finally, according to the area of residence, children from urban areas showed greater association with having higher obesity parameter (except for WC) without showing statistically significant differences for any obesity parameter. Data on absolute and relative frequencies is available in Appendix A.

Figure 2 indicates the association between different status in the anthropometric categorical variables of the study and the age group, sex, type of schooling, per capita income, area of residence and immigrant status. Therefore, no statistically significant association was shown in the case of BMI, WC, and WHtR according to age group and sex. Conversely, a higher association of excess weight was found in children from public schools, as well as of having abdominal obesity. Lower association of having an inadequate %BF were found in the participants aged 6–9 and in boys, in those from private schools, in the low per capita income group, in children from rural areas, and in native students (only statistical significance in the case of age group and immigration). Information on odds ratio and 95% confident intervals is presented in Appendix A.

## 4. Discussion

The aim of the present study was to determine, compare, and examine the influence of age, sex, type of the schooling, per capita income, area of residence, and immigrant status on obesity parameters in schoolchildren aged 6–13 from the Region of Murcia. In this line, this study revealed that a great number of schoolchildren are overweight/obese in the analysed sample. Our results are higher than those reported by the last Estudio ALADINO [22], which reported a prevalence of excess weight of 41.3% for boys and 39.7% for girls, and by the Estudio PASOS [35], which reported an 34.9% of excess weight. According to the Estudio ALADINO (2019), the obesity trends seem to be stabilising in Spain. However, despite this apparent stabilisation, Spain has been pointed out as the European country with the highest prevalence rate of obesity in children (17.7%) in the latest data (2015–2017) from the Childhood Obesity Surveillance Initiative (COSI) led by the WHO [2]. One possible reason for this higher prevalence could be the fact that the municipality with greater per capita income of this study was slightly over 20,000€ and it has been recently shown that the prevalence of obesity is significantly greater in schoolchildren with family incomes below €18,000 per year [22]. Also, a further explanation could be that one out of three children is at risk of child poverty and social exclusion in the Region of Murcia [27], which could at least partially explain these superior figures. However, we must consider that, as reported in the scientific literature, the etiology of obesity is multifactorial [3], so other factors could influence this greater prevalence.

In addition, in relation to age, our adjusted model did not show statistical significance in order to have excess weight. Obesity prevalence rises as children grow older [36], and age is the single major predictor of obesity in children [37]. However, it was remarkable that we did not observe any association between obesity (assessed by BMI) and age in contrast to the findings of other authors who reported that obesity significantly increased with age [38,39]. This fact was not verified in the case of %BF, which indicated higher prevalence of older children (aged 10–13) with high adiposity. These differences among methods could be explained by the different procedures that they use to examine the obesity levels. For example, BMI is not able to differentiate the amount of fat mass and free-fat mass to body weight and could cause the wrong identification of obesity status when used in children [40]. At the same time, the so-called “adiposity rebound” phenomenon in the scientific literature, characterised by an increase in body fat at around the age of 8, continues until the end of the growth process [41].

Concerning sex, the categorisation of BMI reported very close prevalence of excess weight between boys and girls. Thus, we observed that being a girl or boy did not predict the likelihood of having excess weight/obesity, a finding that is consistent with some previous studies conducted [42,43]. However, contrary to our results, some other studies either reported differences in obesity prevalence between the sexes [17,21,44] or found girls to be at increased risk for childhood obesity [39]. In the case of %BF, we found a greater number of girls with higher adiposity than boys. Our results agree with other previous studies [45,46] and was able to verify the long differences well-known for %BF [47]. Nevertheless, we must consider that we used the Siri’s equation [33] that measures four skinfolds of upper body (biceps, triceps, subscapular, and suprailiac) without considering the lower body, which could modify this results. At the same time, another aspect that could have influenced this is the fact that the Siri’s equation [33] seems to overestimate the %BF between boys and girls, according to some studies [46,48].

Regarding the type of schooling, children from public school were more likely to have excess weight when compared to children of private schools. This fact matches with the findings of other studies which evaluated children and adolescent’s excess weight in different countries [49,50] and in Spain [17]. Therefore, it must be remarkable that public school children were more than three times likely to have abdominal obesity. These results seem to point out to the remarkable difference between public and private schools. This might be due to the fact that, in Spain, public schools are funded by the government, while private schools can charge considerably higher fees (tuition, uniform, or school materials). For this reason, only parents who can afford the fees would choose to enrol their children in private schools; the influence of socioeconomic factors in childhood obesity has already been indicated in Spain [22]. At the same time, families with low socioeconomic status may have less access or are not able to afford healthy foods and beverages compared to wealthier homes [51].

Conversely, a higher prevalence of excess weight, abdominal obesity, and especially of high trunk fat mass were shown in children from high per capita income. Our findings match with those reported by Arruda et al. [52], who reported a greater proportion of excess weight among children from families with more favourable living environments (e.g., greater per capita income). Similarly, Beal et al. [20] have recently found that higher per capita income was linked to a greater prevalence of childhood excess weight in Vietnam. Thus, income inequality could affect childhood obesity, through an increase in excessive food intake, as well as in a more sedentary lifestyle which could favour the development of overweight/obesity [1,53,54]. However, caution is needed when interpreting the results because of certain aspects. First, per capita income is referred to the municipality and not to the individual’s socioeconomic status. Second, these authors used different criteria to determine per capita income, as well as different cut-off points. Third, the per capita income indicator may vary depending on the analysed country and presents certain limitations to describe wealth inequalities (e.g., gross national income or Gini index) [53].

Concerning the area of residence, the evidence available on this relationship is inconclusive [26]. For example, Bel-Serrat et al. [26] indicated no association with obesity in a large and representative sample of Irish children aged 8–12. In a systematic review of Chinese studies, Guo et al. [18] showed that the prevalence of excess weight was greater in urban areas. Conversely, other studies have indicated that children living in cities with less density of population present higher rates of childhood obesity than children who reside in urban areas, both in North America [39] and Europe [54]. Moreover, one study carried out by Vaquero-Álvarez et al. [23] found a prevalence of 48.5% of excess weight in a sample of Spanish children from rural areas. One explanation could lie in the fact that, although it is thought that a rural life implies physically demanding activities, it is not always the case, and it could have an influence in the greater prevalence of obesity in rural areas [55]. Despite the above, we found higher obesity prevalence in children from urban areas. This fact could be explained (at least in part) because the analysed schools in our study belonged to adjacent cities, so there might not be much differences between the schoolchildren. Furthermore, the total population reached in the city with the largest number of inhabitants was only around 20,000 inhabitants. These aspects, together with the criterion used for the categorisation of the area of residence [28], could have influenced the results obtained. Moreover, obesity rates by area of residence have not yet been well studied in depth [39].

In the case of immigration, we observed that immigrant children had a higher prevalence of excess weight and high adiposity. This fact is consistent with others studies performed both in Europe [56] and Spain [16,57], suggesting that belonging to an immigrant family has a higher prevalence of excess of weight. In this sense, immigrant children are more likely to suffer from social exclusion, which seems to lead to unhealthy behaviour [58] and probably to increased body weight. However, it has been pointed out that the level of adiposity may differ due to the level of integration of the local culture rather than the fact of being an immigrant itself [59]. On the other hand, the recent study by de Bont et al. [19] also found lower figures among children of African and Asian nationalities. Moreover, these same authors showed that the risk of having excess weight increased over time, according to the cumulative ages of residence in Spain. This tendency has been identified in other studies, in which it was considered that exposure and acculturation to the Western lifestyle among the immigrant population led to this increase [60]. Nonetheless, data on childhood obesity in immigrant population in Europe and Spain are limited and may fluctuate considerably in relation to what is described in the United States [19].

In this study, we found some strengths that characterise our study, we assessed several obesity-related parameters such as WC, WHtR, or %BF, apart from the most commonly used BMI. This choice is based on the intention to offer a better understanding of the childhood obesity problem since, according to previously recommendations, BMI should be complemented by other anthropometric parameters (WC, for instance) to establish efficiently the risk in individuals [14,15]. Furthermore, the inclusion of several sociodemographic factors resulted in two essential aspects: on the one hand, a better understanding of the influence of these factors on the differences in childhood obesity in the region of Spain with the highest number of obese children [27]; on the other hand, a call for both regional and national government institutions to assume responsibility, leadership, political commitment, and take action in line with the recommendations of the WHO to address childhood obesity in the long term [5]. Contrariwise, there are certain limitations that must be declared. First, because of the cross-sectional design of this study, we are not able to establish cause-effect associations. Another limitation that we found is related to the lack of information on the individual’s developmental stage—for example, Tanner Stage, as an indicator of sexual maturation. In this line, differences in the distribution of BF begin at the age of the puberty, with boys developing a distribution which favours central deposition of fat, irrespective of their total BF [47]. Moreover, although we reported on per capita income of the different cities, information on the individual’s socioeconomic status could provide more accurate information about this variable.

Finally, our study suggested that overweight/obesity among schoolchildren in the Region of Murcia is higher than the overall prevalence of Spain. The result of the adjusted analyses showed that age, type of schooling, per capita income, and immigrant status were associated with obesity parameters. The identification of socio-demographic variables related to paediatric obesity can be an appropriate tool for possible early prevention and intervention. Public health policies and interventions aimed at the prevention and treatment of childhood obesity should incorporate a sensitive and clear focus on social and economic inequalities; emphasising on the groups most at risk.

## Figures and Tables

**Figure 1 children-07-00201-f001:**
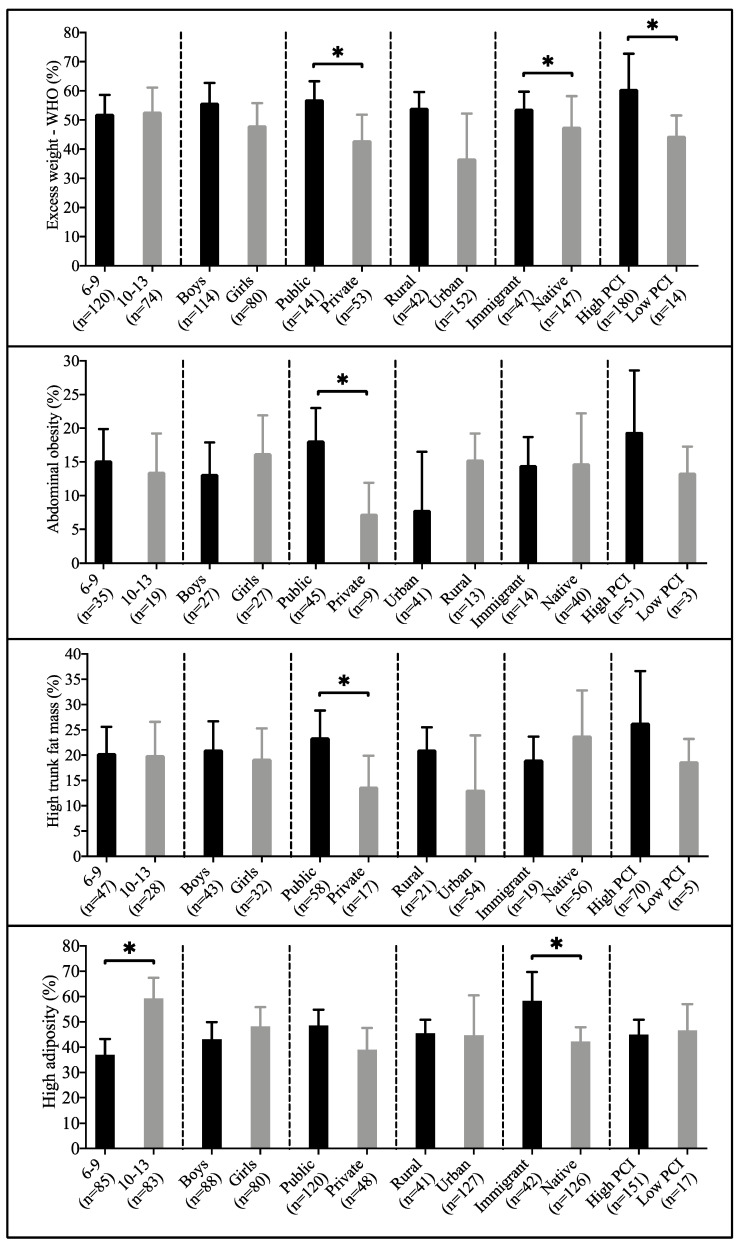
Prevalence of having excess weight, abdominal obesity, high trunk fat mass, and high adiposity according to the different sociodemographic factors. Data presented as number (*n*) and percentage (%). PCI: per capita income. * *p* < 0.050.

**Figure 2 children-07-00201-f002:**
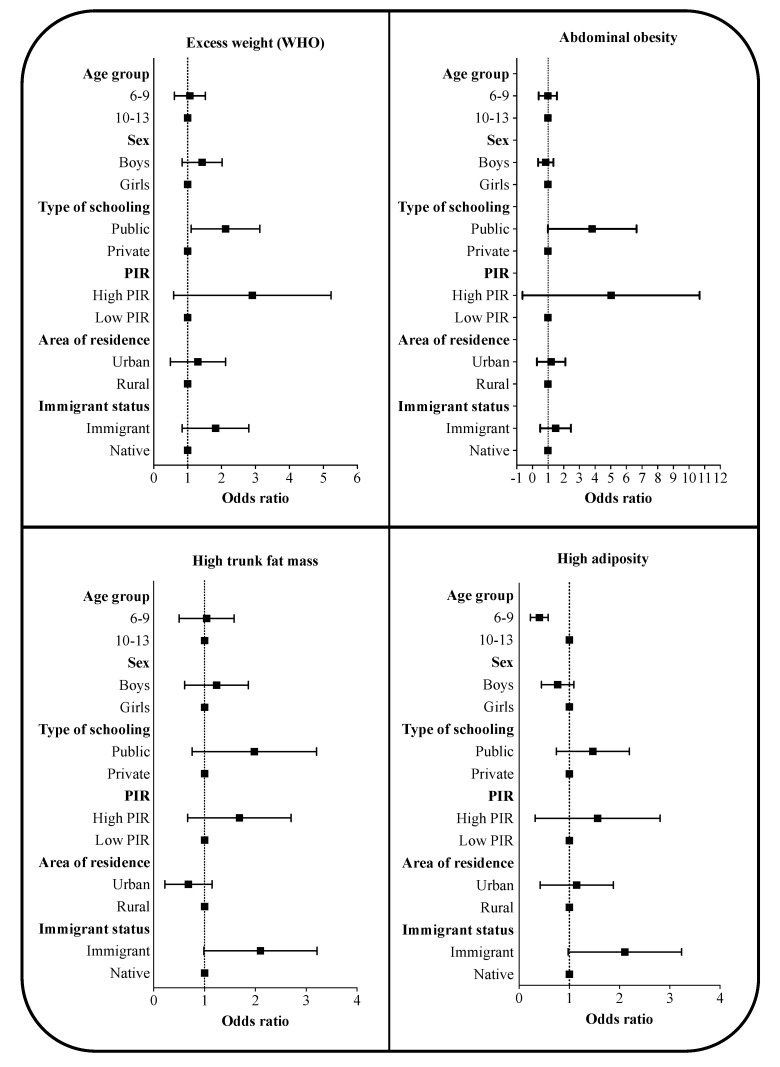
Association of having excess weight, abdominal obesity, high trunk fat mass, and high adiposity according to the different sociodemographic factors. Data expressed as odds ratio (confident intervals 95%). BMI: body mass index; BF: body fat; PIR: per capita income; WHO: World Health Organization; WC: waist circumference; WHtR: waist-to-height ratio. Adjusted by age group, sex, type of schooling, PIR, area of residence, and immigrant status.

**Table 1 children-07-00201-t001:** Characteristics of age and anthropometric parameters of the analysed sample (*n* = 370).

Variables	Mean	SD	CI_95%_
Age (years)	8.7	1.8	(8.5–8.9)
Weight (kg)	35.7	10.9	(34.6–36.8)
Height (cm)	1.36	0.12	(1.35–1.37)
BMI (kg/m^2^)	19.05	3.71	(18.67–19.43)
WC (cm)	62.1	8.2	(61.3–62.9)
WHtR (WC/Height (cm))	0.46	0.05	(0.45–0.47)
BF (%)	29.63	7.13	(28.90–30.36)
BF (kg)	11.17	5.93	(10.56–11.78)
FFM (%)	70.37	7.13	(69.64–71.10)
FFM (kg)	24.54	5.58	(23.97–25.11)

Data expressed as mean, standard deviation and confident intervals (95%). BMI: body mass index; BF (%): Body fat percentage; BF (kg) = Body fat in kilograms; FFM: free-fat mass; WC: waist circumference; WHtR: waist-to-height ratio.

**Table 2 children-07-00201-t002:** Prevalence of different obesity parameters in the analysed simple.

Variables	*n*	%	CI_95%_
BMI (WHO)			
Overweight	104	28.1	(23.8–32.9)
Obesity	90	24.3	(20.2–29.0)
Excess weight	194	52.4	(47.3–57.5)
WHtR			
Abdominal obesity	54	14.6	(11.4–18.6)
WC			
High trunk fat mass	75	20.3	(16.5–24.7)
%BF			
High adiposity	168	45.4	(40.4–50.5)

Data expressed as frequencies, percentages and confident intervals (95%). BMI: body mass index; BF: body fat; WHO: World Health Organization; WC: waist circumference; WHtR: waist-to-height ratio.

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
