# Peer review of "Sociodemographic Correlates of Obesity among Spanish Schoolchildren: A Cross-Sectional Study"

_children, 2020, doi:10.3390/children7110201_

Round 1

Reviewer 1 Report

The authors present findings from a descriptive study, regarding the prevalence of excess weight in children residing a Spanish county, using multiple auxological indices. While the study has some interest, many points should be taken into account

General comments: The article requires major language editing

Abstract: please provide the full wording prior to the abbreviated form (i.e. WOF-IOTF). Also, please change to CI95% to 95% CI (in the abstract, as well as the whole document)

Introduction: the introduction is clear and concise. However, the authors could provide a more justified explanation as to why this very specific study may add to intrnational literature. For instance, please explain specific socioeconomic influences that may exist, a possible excess weight gradient in the country etc. This should also be in line with the aim of the study

Methods: could the authors please provide (if they recorded these data) the response rate of the study, and the basis of selection of the 6 primary schools.

The definition of immigration is a bit confusing, specifically in the case of children with one parent not being Spanish. Could the authors please provide justification for this definition, or the rationale behind it? Is this definition supported by literature?

Table 2 & 3: please consider restructuring the Tables, so as to avoid presenting self-explanatory variables. For instance, the no excess weight group can be ommited, as well as the adequate adiposity, no abdominal obesity etc.

L170: please provide a rough description of these studies, as well as the prevalence of pooled overweight they recorded

L172: the statement is a bit confusing. It cannot be that while the rates of obesity are reaching a plateau, Spain has also the highest prevalence of obesity in Europe?

L178: what is BMC?

L188-199: please avoid using male and female, and instead use boy and girl. The first indicate the gender of the participants, whereas in the methods section you state that the sex of the participants was reported (not the gender).

Discussion: After presenattion of the results, are there vast differences in the county compared to the rest of the country? The authros should more thoroughly discuss their findings compare to both Spanish, and international literature. Then, the authors should provide explanations for the possible differences that exist in their setting with the rest of Spain (in line with the measures acquired). Could this finding provide a new paradigm for pediatric obesity? Are novel population groups at risk highlighted etc

Reviewer 2 Report

The purpose of this ms is to report relationships between selected demographic variables and obesity parameters in schoolchildren living in the Region of Murcia, Spain. Comments appear below.AbstractNo concernsIntroductionNo evidence is given for the examination of the selected demographic characteristics; please provide evidence to support the decision to examine the relationship between the selected demographic characteristics and obesity, as well as the rationale for selecting these particular obesity parameters – i.e., what is known from the literature about the most useful/representative parameters in general – suggest not limiting response to just studies conducted with children in Spain; the broader literature may offer important insightsMaterials and MethodsWhat was the justification for using the selected sample; for example, was it a convenience sample, or was their selection driven by a particular criterion (i.e., SES, large population of immigrant students, etc)Please clarify what specific students were included – it sounded as though the study only included students in PE classes; if yes, could children who did not participate in PE classes be potentially different in terms of obesity status?Over what time period were data collected?What were the inclusionary criteria for inclusion in the study?ResultsPlease identify the total number of students who completed assessments, and the number of students included in the analysesDiscussionLimitations – expand to include inclusion of only children in PE class if this is the case – see rationale above in comment under Discussion headingReferencesNo concerns; many were published during the past decadeFigures/TablesAdd sample size to tables.

Round 2

Reviewer 1 Report

The quality of the manuscript has increased. Beyond extensive editing in English, no further comments are made

Author Response

Thank you so much for both your comments and review. We have had the manuscript reworked by a English translator to improve the possible mistakes.

Kind regards,

Reviewer 2 Report

The authors did an adequate job responding to my previous comments. However, there is a lingering concern that this research is a reiteration of what's already known.  I am unclear what additional information this work contributes to the literature.  Perhaps the authors can emphasize the unique contribution of this work in the discussion.

Also, on lines 231-2, I suggest  deleting this statement.  It could be interpreted as 'blaming the victim', which I do not believe is the intention of the authors.

Author Response

Thank you very much for your feedback. According to your suggestion, the lines 231-232 have been removed. On the other hand, we have tried to give a better justification of our study as follows (Lines 280-284): "Furthermore, the inclusion of several sociodemographic factors allowed two essential aspects: on the one hand, a better understanding of the influence of these factors on the differences in childhood obesity in the region of Spain with the highest numbers of obese children [27]; on the other hand, to call on government institutions to take responsibility, leadership and political commitment, by taking action in line with the recommendations of the WHO to tackle childhood obesity in the long term [5]."

Kind regards,